# SHAPE OR TEXTURE: UNDERSTANDING DISCRIMINATIVE FEATURES IN CNNs

**Md Amirul Islam**[1,6]     **Matthew Kowal**[1]     **Patrick Esser**[3]     **Sen Jia**[2]
**Björn Ommer**[3]     **Konstantinos G. Derpanis**[1,5,6]     **Neil D. B. Bruce**[4,6]
[1]Department of Computer Science, Ryerson University, Canada
[2]University of Waterloo, Canada
[3]IWR, HCI, Heidelberg University, Germany
[4]School of Computer Science, University of Guelph, Canada
[5]Samsung AI Centre Toronto, Canada
[6]Vector Institute for AI, Canada
{mdamirul.islam,matthew.kowal,kosta}@ryerson.ca, sen.jia@uwaterloo.ca
{patrick.esser,björn.ommer}@iwr.uni-heidelberg.de, brucen@uoguelph.ca

## ABSTRACT

Contrasting the previous evidence that neurons in the later layers of a Convolutional Neural Network (CNN) respond to complex object shapes, recent studies have shown that CNNs actually exhibit a 'texture bias': given an image with both texture and shape cues (e.g., a stylized image), a CNN is biased towards predicting the category corresponding to the texture. However, these previous studies conduct experiments on the *final classification output* of the network, and fail to robustly evaluate the bias contained (i) in the latent representations, and (ii) on a per-pixel level. In this paper, we design a series of experiments that overcome these issues. We do this with the goal of better understanding what type of shape information contained in the network is discriminative, where shape information is encoded, as well as when the network learns about object shape during training. We show that a network learns the majority of overall shape information at the first few epochs of training and that this information is largely encoded in the last few layers of a CNN. Finally, we show that the encoding of shape does not imply the encoding of localized per-pixel semantic information. The experimental results and findings provide a more accurate understanding of the behaviour of current CNNs, thus helping to inform future design choices.

## 1 INTRODUCTION

Convolutional neural networks (CNNs) have achieved unprecedented performance in various computer vision tasks, such as image classification (Krizhevsky et al., 2012; Simonyan & Zisserman, 2015; He et al., 2016), object detection (Ren et al., 2015; He et al., 2017) and semantic segmentation (Long et al., 2015; Chen et al., 2017; Islam et al., 2017). Despite their black box nature, various studies have shown that early layers in CNNs activate for low-level patterns, like edges and blobs, while deeper layers activate for more complex and high-level patterns (Zeiler & Fergus, 2014; Springenberg et al., 2014). The intuition is that this hierarchical learning of latent representations allows CNNs to recognize complex object shapes to correctly classify images (Kriegeskorte, 2015). In contrast, recent works (Brendel & Bethge, 2019; Hermann & Lampinen, 2020) have argued that CNNs trained on ImageNet (IN) (Deng et al., 2009) classify images mainly according to their *texture*, rather than object *shape*. These conflicting results have large implications for the field of computer vision as it suggests that CNNs trained for image classification might be making decisions based largely off spurious correlations rather than a full understanding of different object categories.

One example of these spurious correlations is how the Inception CNN (Szegedy et al., 2015) recognizes the difference between 'Wolf' and 'Husky', based on whether there is snow in the background (Tulio Ribeiro et al., 2016). Recognizing object shapes is important for the generalization to out-of-domain examples (e.g., few-shot learning), as shape is more discriminative than texture when

texture-affecting phenomena arise, such as lighting, shading, weather, motion blur, or when switching between synthetic and real data. In addition to performance, identifying the discriminative features that CNNs use for decision making is critical for the transparency and further improvements of computer vision models. While the model may achieve good performance for a certain task, it cannot communicate to the user about the *reasons* it made certain predictions. In other words, successful mod-

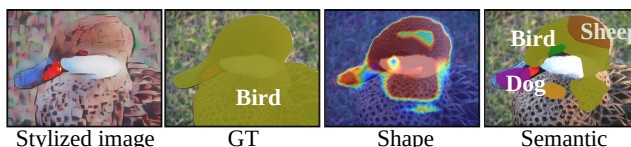

Stylized image    GT    Shape    Semantic

Figure 1: A shape biased model (trained on Stylized ImageNet) makes predictions based on the object's *shape*, or does it? Extracting binary ($3^{rd}$ column) and semantic ($4^{th}$ col.) segmentation maps with a one convolutional layer readout module shows that, while the model classifies the image level shape label correctly as a 'bird', it fails to encode the full object *shape* ($3^{rd}$ col.) as well as fails to categorically assign every object pixel to the 'bird' class ($4^{th}$ col.).

els need to be good, and interpretable (Lipton, 2019). This is crucial for many domains where causal mechanisms should play a significant role in short or long-term decision making such as healthcare (e.g., what in the MRI indicates a patient has cancer?). Additionally, if researchers intend for their algorithms to be deployed, there must be a certain degree of *trust* in the decision making algorithm.

One downside of the increasing abstraction capabilities of deep CNNs is the lack of interpretability of the latent representations due to hidden layer activations coding semantic concepts in a distributed fashion (Fong & Vedaldi, 2018). It has therefore been difficult to precisely quantify the type of information contained in the latent representations of CNNs. Some methods have looked at ways to analyze the latent representations of CNNs on a neuron-to-neuron level. For instance, (Bau et al., 2017) quantify the number of interpretable neurons for a CNN by evaluating the semantic segmentation performance of an individual neuron from an upsampled latent representation. Later work (Fong & Vedaldi, 2018) then removed the assumption that each neuron encodes a single semantic concept. These works successfully quantify the number of filters that recognize textures or specific objects in a CNN, but do not identify *shape* information within these representations.

The most similar works to ours are those that aim to directly quantify the shape information in CNNs. For example, (Geirhos et al., 2018) analyzed the outputs of CNNs on images with conflicting shape and texture cues. By using image stylization (Huang & Belongie, 2017), they generated the Stylized ImageNet dataset (SIN), where each image has an associated shape *and* texture label. They then measured the 'shape bias' and 'texture bias' of a CNN by calculating the percentage of images a CNN predicts as either the shape or texture label, respectively. They conclude that CNNs are 'texture biased' and make predictions mainly from texture in an image. This metric has been used in subsequent work exploring shape and texture bias in CNNs (Hermann & Kornblith, 2019); however, the method only compares the *output* of a CNN, and fails to robustly quantify the amount of shape information contained in the *latent representations* (note that they refer to 'shape' as the entire 3D form of an object, including contours that are not part of the silhouette, while in our work, we define 'shape' as the 2D class-agnostic silhouette of an object). Thus, the method from (Hermann & Kornblith, 2019) cannot answer a question of focus in our paper: 'What fraction of the object's shape is actually encoded in the latent representation?'. Further, as their metric for *shape* relies solely on the semantic class label, it precludes them from evaluating the encoded shape and associated categorical information on a per-pixel level. For instance, we show in Fig. 1 that shape biased models (i.e., trained on stylized images) do not classify images based on the entire object shape: even though the CNN correctly classifies the image as a bird, only the *partial* binary mask (i.e., 'shape') can be extracted from the latent representations and it cannot attribute the correct class label to the *entire object region* (i.e., semantic segmentation mask).

**Contributions.** To address these issues, we perform an empirical study on the ability of CNNs to encode shape information on a neuron-to-neuron and per-pixel level. To quantify these two aspects, we first approximate the mutual information of latent representations between pairs of semantically related images which allows us to estimate the number of dimensions in the feature space dedicated to encoding shape and texture. We then propose a simple strategy to evaluate the amount of shape information contained in the internal representations of a CNN, on a per-pixel level. The latter technique is utilized to distinguish the quality of different shape encodings, regardless of the number of neurons used in each encoding. After showing the efficacy of the two methods, we reveal a number of meaningful properties of CNNs with respect to their ability to encode shape information,

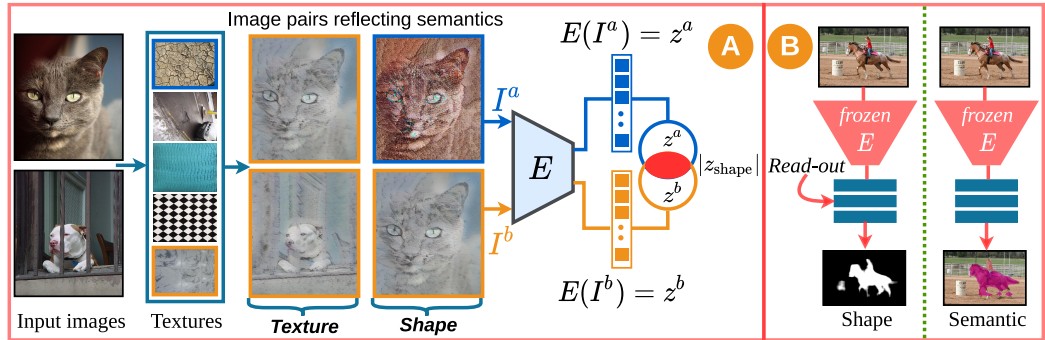

Figure 2: Illustration of the techniques used to quantify shape in this paper. **(A)** Estimating the dimensionality of semantic concepts in latent representations: We stylize each image with five textures to generate image pairs which share the semantic concepts *shape* (right pair) and *texture* (left pair). We feed these image pairs (shown is *shape*) to an encoder, $E(\cdot)$, and calculate the mutual information between the two latent representations, $z^a$ and $z^b$, to estimate the dimensionality, $|z_{\text{shape}}|$. **(B)** We quantify the shape information encoded in a convolutional neural network by freezing the weights, and then training a small read-out module (i.e., three $3{\times}3$ convolutional layers) on the latent representation to predict either a binary or semantic segmentation map.

including the following: **(i)** Biasing a CNN towards shape predominantly changes the number of shape encoding neurons in the last feature encoding stage. **(ii)** When a CNN is trained on ImageNet, the majority of shape information is learned during the first few epochs. **(iii)** A significant amount of shape is encoded in the early layers of CNNs, which can be utilized to extract additional shape information from the network, by combining with shape encodings from deeper layers. **(iv)** Encoding the shape and class of an object does not imply the useful encoding of localized per-pixel categorical information. All code will be released to reproduce data and results.

## 2   DO CNNS SPEND MORE LEARNING CAPACITY ON SHAPE OR TEXTURE?

With the goal of revealing the characteristics of *where*, *when*, and *how much* shape information is encoded in CNNs, we first aim to quantify the number of dimensions which encode shape in a CNN's latent representation. This analysis on the latent representations will allow us to determine *where* the network spends learning capacity on shape, while other methods that focus solely on the network outputs have difficulty measuring the difference in shape information between convolutional layers.

### 2.1   ESTIMATING SHAPE AND TEXTURE DIMENSIONALITY

Previous works (Bau et al., 2017; Esser et al., 2020) proposed various mechanisms to reveal the semantic concepts encoded in latent representations of CNNs. To quantify the amount of texture and shape information, we follow the approach of (Esser et al., 2020), where the number of neurons that represent a certain semantic concept is estimated. Given a pretrained CNN encoder, $E(I) = z$, where $z$ is a latent representation, we aim to estimate the dimensionality of the semantic concepts shape and texture within $z$. The main idea is that the mutual information between image pairs, $I^a$ and $I^b$, which are similar in a semantic concept, will be preserved in a neuron $z_i$ only if the neuron encodes that specific semantic concept. Hence, the mutual information between the corresponding neuron pairs, $z_i^a = E(I^a)$ and $z_i^b = E(I^b)$, can be used to quantify the degree to which a semantic concept is represented by the neuron. A simple and efficient estimate for their mutual information $\text{MI}(z_i^a, z_i^b)$ can be obtained based on the correlation coefficient $\rho_i$. Indeed, under the assumption that the marginal distribution of the neuron $z_i$ is Gaussian, the correlation coefficient $\rho_i$ provides a lower bound on the true mutual information through the following relationship which becomes tight for jointly Gaussian $z_i^a, z_i^b$ (Kraskov et al., 2004; Foster & Grassberger, 2011).

$$\text{MI}(z_i^a, z_i^b) \geq -\frac{1}{2}\log(1 - \rho_i^2), \quad \text{where } \rho_i = \frac{\text{Cov}\big(z_i^a, z_i^b\big)}{\sqrt{\text{Var}(z_i^a)\,\text{Var}(z_i^b)}}. \tag{1}$$

To quantify how well a concept $k$ is represented in terms of the number of neurons $|z_k|$ that encode the concept, we compute a score for each concept and the relative number of neurons is determined

Table 1: **Dimensionality estimation** of semantic factors $|z_k|$ for the stage-5 latent representation. Note that the total dimension of the latent representation, $|z|$, is 2048 for all networks, and that the remaining dimensions are allocated to the 'residual' factor. (a) ResNet50 compared with BagNets. BagNets have more neurons which encode *texture* than *shape* due to their restricted receptive field. (b) Networks with varying levels of shape bias. The number of neurons which encode shape correlates with shape bias. (c) Deeper networks contain more *shape* encoding neurons.

| Model | Factor $|z_k|$ | |
|---|---|---|
| | Shape | Texture |
| ResNet50 | **349** | 692 |
| BagNet33 | 284 | 825 |
| BagNet17 | 278 | 839 |
| BagNet9 | 276 | **841** |

| Training Data | Factor $|z_k|$ | |
|---|---|---|
| | Shape | Texture |
| IN | 349 | **692** |
| SIN | **536** | 477 |
| (SIN+IN)→IN | 376 | 640 |

| Model | Factor $|z_k|$ | |
|---|---|---|
| | Shape | Texture |
| ResNet50 | 349 | **692** |
| ResNet101 | 365 | 667 |
| ResNet152 | **371** | 661 |

with a softmax over these scores and a baseline score. The latter is given by the number of neurons $|z|$, and shape and texture scores are given by the sum of their respective correlation coefficients $\rho_i^{\text{shape}}$ and $\rho_i^{\text{texture}}$, which are computed according to Eq. 1 with statistics taken over image pairs that are similar in shape and texture, respectively. Note that $k \in \{1, 2\}$ in our case, and the remaining dimensions not captured in any of the two semantic factors are allocated to the *residual* semantic factor, which by definition captures all other variability in the latent representation, $z$.

**Stylized PASCAL VOC 2012 Dataset.** Our goal is to estimate the dimensionality of two semantic concepts: (i) *shape* and (ii) *texture*, and analyze pixel-wise shape information. Therefore we must generate a dataset that we can sample image pairs which share the semantic factors shape or texture, and have per-pixel object annotations. To accomplish this goal, we create the *Stylized PASCAL VOC 2012* (SVOC) dataset. Similar to SIN, we use the AdaIN style transfer algorithm (Huang & Belongie, 2017) to generate stylized images from the PASCAL VOC 2012 dataset (Everingham et al., 2010) with the same settings and hyperparameters as in the original paper (Huang & Belongie, 2017). We choose five random textures from the Describable Textures Dataset (Cimpoi et al., 2014) as the styles and we stylize every PASCAL VOC image with all five of these textures. For a fair comparison with models trained on ImageNet variants, we take only the images from PASCAL VOC which contain a single object. With the SVOC dataset, we can now sample image pairs which are similar in *texture*, by using two images from different categories but stylized with the same texture (Fig. 2(A) left), or *shape*, by using the same image stylized with two different textures (Fig. 2(A) right).

## 2.2 RESULTS

We now evaluate the efficacy of the dimensionality estimation method by comparing two networks which differ significantly in their ability to encode shape information. The first is a standard ResNet50 architecture and the second is the recently proposed BagNet (Brendel & Bethge, 2019). BagNets are a modified version of ResNet50 that restrict the height and width of the effective receptive field of the CNN to be a fixed maximum, i.e., either 9, 17, or 33 pixels. This patch-based construction precludes BagNets from classifying images based on extended shape cues. The results of this comparison are presented in Table 1(a) where both the ResNet50 and the BagNet variants are trained on IN. Note that 'Stage' refers to a residual block in a ResNet (i.e., there are five stages in ResNet) and all experiments in Table 1 use the stage-5 features. As expected, BagNets have more neurons encoding texture than the ResNet50 and there is a clear correlation between the receptive field of the network and the amount of shape encoded. As the receptive field decreases, the number of neurons encoding texture increases even further, while the number of neurons encoding shape decreases.

We now examine whether the 'shape bias' metric (Geirhos et al., 2018) correlates with the number of shape encoding neurons. Table 1(b) compares the estimated dimensionality of a ResNet50 trained on ImageNet against networks which are biased towards shape using two different training strategies: (i) training solely on SIN and (ii) training on SIN and IN simultaneously followed by fine-tuning on IN (denoted as (SIN+IN)→IN, which achieves the best accuracy on ImageNet top-1% out of the three variations (Geirhos et al., 2018)). ResNet50 trained on IN has far more neurons dedicated to encoding texture than shape. There is a large difference when training a ResNet solely on SIN, where it has less neurons which encode texture than shape. When trained and fine-tuned on (SIN+IN) and IN, respectively, there is an increase in the number of neurons which encode shape compared to IN. We consider if there is any pattern in the number of neurons encoding shape or texture as the network depth increases.

As can be seen in Table 1(c), networks have more shape and less texture neurons as depth increases. This may be due to the increase in learning capacity of the deeper networks, as more hierarchical representations allow for the network to learn increasingly complex shapes compared to the shallower networks. Further, deeper networks have stronger long range connections due to a larger effective receptive field potentially resulting in additional shape encoding neurons. The increase in shape understanding could be one of the reasons why deeper networks achieve better performance on various tasks, e.g., image classification. Finally, we assess the consistency between the dimensionality estimation technique and network dissection (Bau et al., 2017), another method which estimates the number of neurons representing different concepts (described in Sec. A.2). Since network dissection cannot estimate shape dimensionalities, the comparison is limited to the texture dimensions shown in Table 2. Except for the case of BagNet9 and a difference in the absolute numbers of neurons (discussed in Sec. A.2), both methods agree about the correlation between texture dimensionality and the receptive field, which provides further evidence that dimensionality estimates quantify the relevance of semantic concepts faithfully.

Table 2: Comparing dimensionality estimations for the *texture* factor between two methods. Both methods show that more texture neurons are found in representations with smaller receptive fields.

| Method | Res152 | Res101 | Res50 | Bag33 | Bag17 | Bag9 |
|---|---|---|---|---|---|---|
| **Net. Diss.** | 433 | 481 | 499 | 592 | 623 | 537 |
| **Dim. est.** | 661 | 667 | 692 | 825 | 839 | 841 |

**Stage-Wise Analysis of Shape and Texture Dimensionality.** We now explore *where* CNNs encode shape by applying the dimensionality estimation method with the latent representations from ResNet50 stages one to five with different amounts of shape bias. Due to the different dimensions at each of these stages, we present the results as the percentage of dimensions encoding the particular semantic factor, $|z_k|/|z|$, where $|z|$ refers to length of the latent representation. Table 3 shows that *all stages of the network* encode shape with an increase in the last two stages. Further, biasing the models towards shape only changes the percentage of shape encoding in the final two stages. Beginning at the fourth stage, there is a significant jump in the number of shape dimensions for all three models with the shape biased models having a larger increase. At the final stage, latent representations encode even more shape, where SIN in particular has a large increase of 8.5%. This indicates that biasing a model towards shape mainly affects the last two stages of the network, suggesting that future work could focus on improving the shape bias of earlier layers. An increase in shape dimensions is *inversely proportional* to the amount of texture dimensions. Notably, from stage four to stage five, there is a large drop in the amount of texture dimensions for all networks.

Table 3: Percentage of neurons ($|z_k|/|z|$) encoding different semantic concepts, $k$, for different stages of ResNet50 trained for various levels of shape bias. While a moderate percentage of neurons encode shape in stages $f_1$, $f_2$, and $f_3$, the majority of shape neurons are found in stage $f_5$. Networks with shape bias learn additional shape information in stage $f_5$.

| Stage | IN Factor $|z_k|/|z|$ | | SIN Factor $|z_k|/|z|$ | | (SIN+IN)→IN Factor $|z_k|/|z|$ | |
|---|---|---|---|---|---|---|
| | Shape | Texture | Shape | Texture | Shape | Texture |
| $f_1$ | 12.5% | 42.2% | 12.5% | 42.2% | 12.5% | 42.2% |
| $f_2$ | 14.1% | 40.2% | 14.1% | 40.6% | 14.1% | 40.6% |
| $f_3$ | 14.6% | 39.5% | 14.8% | 39.5% | 14.8% | 39.5% |
| $f_4$ | 15.3% | 37.9% | 17.7% | 34.7% | 17.7% | 34.8% |
| $f_5$ | 17.0% | 33.8% | 26.2% | 23.3% | 18.4% | 31.2% |

**When Does Shape Become Relevant During Training?** To answer the question 'When do models learn to encode shape and texture during training?', we capture the changes of *shape* and *texture* occurring over the course of training a classifier on ImageNet (IN) and Stylized ImageNet (SIN). We obtain 18 different instances of a ResNet50 model during training on IN and SIN, each representing a checkpoint between epochs 0 and 90 (equally distributed). For each checkpoint, we measure the dimensionality of shape and texture semantic factors and plot the results in Fig. 3. The shape factor in the stage-5 latent representations for both IN (Fig. 3 middle) and SIN (Fig. 3 right) models become increasingly more relevant during the course of training, however the percentage of dimensions grows much larger and faster in the case of the SIN trained model. The texture factor decreases as the training progresses in both cases as well. For the stage-4 representation in a model trained on IN (Fig. 3 left), note that the shape encoding neurons increase only marginally over the course of training. This further reveals that a large proportion of shape information is encoded at the deepest layer.

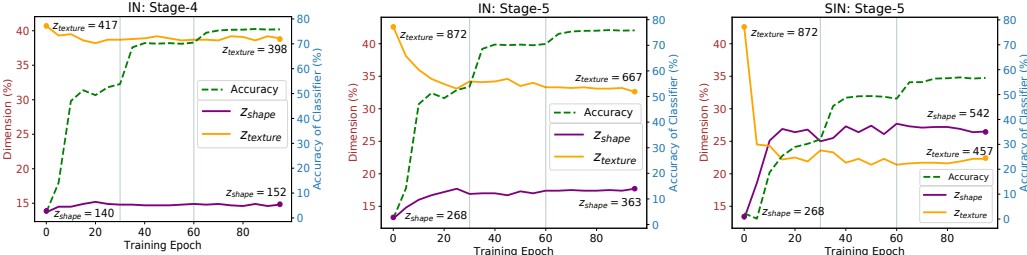

Figure 3: Analyzing the number of dimensions in a ResNet50 which encode shape ($|z_{shape}|$) and texture ($|z_{texture}|$) over the course of ImageNet (IN, left two) and Stylized ImageNet (SIN, right) training. Dimensions are estimated using stage four, $|z^{(4)}| = 1024$, and stage five, $|z^{(5)}| = 2048$, latent representations. When training begins, $z$ is very sensitive to *texture* but over the course of training learns to focus on the *shape* instead (faster in SIN case). The vertical lines represent multiplying the learning rate by a factor of 0.1. Note that the estimated dimensions differ slightly from Table 1 as we trained the IN and SIN models used in this figure from scratch.

## 3 HOW MUCH SHAPE INFORMATION DO CNNS ENCODE?

The previous section measured the dimensionality of shape encodings for various CNNs and settings. We now aim to evaluate the quality of these encodings, and whether more shape encoding neurons implies that more robust shape information can be extracted from these latent representations. We also conduct a set of experiments by targeting the shape and texture-specific neurons (see Sec. 3.3 and Sec. A.1 for results and discussion), revealing an additional link between the two techniques used in our paper (i.e., dimensionality estimation and read-out module). Hermann & Kornblith (2019) measured the quality of shape encodings in a CNN's latent representations by training a linear classifier on the CNN's late-stage features to predict the shape label of SIN images. Quantifying shape information by using image level labels does not allow for the per-pixel evaluation of the encoded shape, and its relation to the associated categorical label, two key components for fully evaluating the characteristics of shape information contained in a particular encoding.

### 3.1 QUANTIFYING SHAPE INFORMATION IN CNN LATENT REPRESENTATIONS

To overcome the aforementioned issues, we consider two tasks which require a detailed understanding of object shape: *binary* and *semantic* segmentation. A 'shape encoding network' (SEN), the network being analyzed, consists of a CNN with fixed weights. We then train a shallow read-out module that takes a latent representation from the SEN, to predict a segmentation map (i.e., binary or semantic). If the read-out module can accurately segment objects with *a binary mask*, we conclude the SEN encodes the precise shape of the objects of interest. Further, the read-out modules ability to perform *semantic* segmentation, measures how much of this encoded shape is successfully localized with per-pixel categorical information. We use ResNet networks of various depths (i.e., 34, 50, and 101) as SENs with a readout module containing either one or three convolution layers with 3×3 kernels.

### 3.2 RESULTS

We use the *trainaug* and *val* split of the VOC 2012 dataset to train and test the read-out module, respectively. The binary segmentation ground truth labels are generated by converting all semantic

Table 4: **Left:** We measure the amount of shape encoded in frozen CNN by training a read-out module on either binary (*Bin*) or semantic segmentation (*Sem*) under different training settings. 'None': random initialization, 'End-to-End': network is not frozen and trained with the read-out module, 'IN': pre-trained on ImageNet. **Right:** Shape information contained in various shape biased models.

| Training | 1 Layer Readout | | | | | | 3 Layers Readout | | | | | |
|---|---|---|---|---|---|---|---|---|---|---|---|---|
| | ResNet34 | | ResNet50 | | ResNet101 | | ResNet34 | | ResNet50 | | ResNet101 | |
| | Bin | Sem | Bin | Sem | Bin | Sem | Bin | Sem | Bin | Sem | Bin | Sem |
| None | 46.5 | 5.2 | 48.0 | 6.1 | 44.9 | 5.1 | 58.0 | 7.2 | 58.0 | 6.0 | 55.0 | 4.8 |
| End-to-End | 80.2 | 63.4 | 80.2 | 62.7 | 81.0 | 65.8 | 82.1 | 67.7 | 82.2 | 68.1 | 82.9 | 71.5 |
| IN | 66.3 | 48.1 | 70.6 | 50.9 | 72.1 | 51.9 | 78.9 | 59.1 | 79.8 | 61.6 | 80.4 | 63.4 |

| Training | ResNet50 | |
|---|---|---|
| | Bin | Sem |
| IN | 79.8 | 61.6 |
| SIN | 76.4 | 53.7 |
| (SIN+IN)→IN | 77.8 | 58.0 |

Table 5: Shape encoding results for different stages of ResNet networks trained on ImageNet. Combining features from early stages increases shape encoding.

| $f_1$ | $f_2$ | $f_3$ | $f_4$ | $f_5$ | ResNet50 | | ResNet101 | |
|---|---|---|---|---|---|---|---|---|
| | | | | | Bin | Sem | Bin | Sem |
| ✓ | | | | | 44.7 | 4.6 | 42.3 | 4.7 |
| | ✓ | | | | 52.7 | 6.4 | 53.0 | 5.6 |
| | | ✓ | | | 59.6 | 10.9 | 57.7 | 9.4 |
| | | | ✓ | | 70.8 | 33.9 | **73.2** | 43.6 |
| | | | | ✓ | **70.6** | 50.9 | 72.1 | **51.9** |
| ✓ | ✓ | | | | 66.0 | 16.6 | 63.8 | 13.5 |
| ✓ | ✓ | ✓ | | | 74.5 | 42.2 | 76.8 | 49.8 |
| ✓ | ✓ | ✓ | ✓ | | **77.3** | **53.7** | **78.2** | **56.2** |
| ✓ | ✓ | ✓ | ✓ | ✓ | **77.3** | 52.9 | **78.2** | 55.2 |

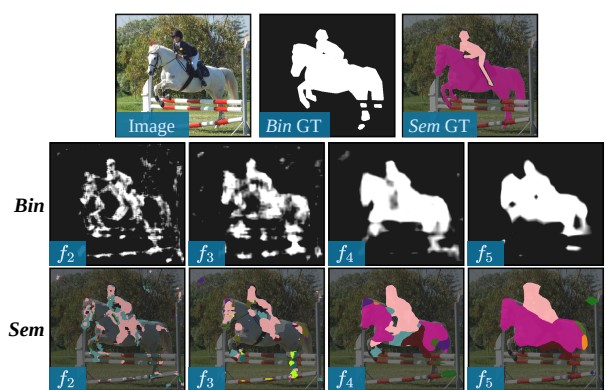

Figure 4: Stage-wise predictions of read-out module on binary (*Bin*) and semantic (*Sem*) segmentation.

categories to a single 'object' class. Note that the binary segmentation and semantic segmentation experiments are done completely independently of one another.

Table 4 presents the results in terms of mean-Intersection-over-Union (mIoU) under different initialization settings with; 'IN': a SEN trained for ImageNet classification, 'None': a SEN with random weight initialization and without any training, 'End-to-End': the SEN and readout module trained in an end-to-end manner on either the binary (*Bin*) or semantic (*Sem*) segmentation ground truth. The None and End-to-End networks represent lower and upper bounds for encoding shape, respectively. All read-out modules in this section are trained on the last layer's latent representations. Interestingly, three convolutional layers can extract similar amounts of shape information from the IN-SEN as the End-to-End-SEN. For example, training the ResNet101 End-to-End-SEN for *Bin* improves the mIoU by merely 2.5% compared to the IN-SEN. ImageNet trained CNNs also contain shape encodings which successfully localize per-pixel categorical information as well, which can be seen when comparing the performance of the IN-SEN and End-to-End SEN, e.g., for ResNet50, the IN-SEN and End-to-End-SEN achieve 61.6% and 68.1%, respectively. This is an interesting result considering the difficulty of semantic segmentation and that *none* of the IN-SEN weights are trained for pixel-wise objectives. Shape information also increases relative to the depth of the network which supports the results presented in Table 1(c). As expected, the End-to-End-SEN and IN-SEN contain significantly more shape information in their latent representations than the baseline None-SEN.

We now evaluate the shape information encoded in networks which have different levels of shape bias. We compare the *Bin* and *Sem* performance of the read-out module trained on the features of three different SENs trained on IN, SIN, and (SIN+IN)→IN. As the validation is on non-stylized images, SIN-SEN has slightly lower performance for *Bin*, and significantly less performance on *Sem*. Such a large difference in performance implies that while the boundary of the object is known, it is difficult for the network to correctly assign per-pixel *categorical* information, a phenomenon further explored in Sec. 3.2.1. Interestingly, the (SIN+IN)→IN-SEN also has slightly lower performance than the IN-SEN for *Bin*, but does not suffer in performance as much as the SIN-SEN in the case of *Sem*.

**Where is Shape Information Stored?** We now examine if the large amount of *shape* information contained in ImageNet pretrained models is equally distributed across different stages of the CNN. In this experiment, we train one layer read-out modules on features from different stages, $(f_1, f_2, f_3, f_4, f_5)$, of the SEN to examine which stage of a CNN encodes shape information. As shown in Table 5, the read-out module trained on the last stage features, $(f_4, f_5)$, achieves higher performance compared to the earlier stage features, $(f_1, f_2, f_3)$, for both *Bin* and *Sem*. This is to be expected, as feature maps from later stages have higher channel dimensions and larger effective receptive fields compared to the feature maps extracted from earlier layers. A surprising amount of shape information (i.e., *Bin*) can be extracted from stages $f_1$, $f_2$ and $f_3$; however, these features lack high-level *semantics* to correlate with this shape information, which can be observed as the corresponding *Sem* performance is much lower. Figure 4 reveals this phenomenon; the *horse* and *person* are outlined even for the early stage binary masks, but are only labelled with correct per-pixel categorical assignments in the later stages. Considering the non-trivial amount of shape information contained in the early stages, we investigate if aggregating multi-stage features encodes more shape compared to the last stage feature, $f_5$. Table 5 (bottom) shows that training a readout module on multi-stage features significantly improves the *Bin* and *Sem* performance, suggesting that tasks

Table 6: Shape encoding results for shape biased ResNet50s on *stylized* VOC12 validation set.

| Training Data | ResNet50 | |
|---|---|---|
| | Bin. | Sem. |
| IN | 45.7 | 8.5 |
| SIN | 60.3 | 26.4 |
| (SIN+IN)→ IN | 44.5 | 9.0 |

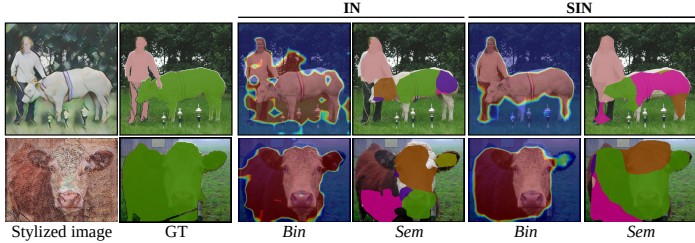

Figure 6: Binary and semantic segmentation masks extracted from CNNs trained on ImageNet (**IN**) and Stylized ImageNet (**SIN**).

requiring shape information may benefit from hypercolumn style architectures (Hariharan et al., 2015). This indicates that some *shape* information is encoded in earlier layers but not captured in the late stages, which agrees with the dimensionality estimation results in Table 3, as around 12.5% and 14% of neurons encode shape in the first stage and second stage, respectively.

**When do CNNs Encode Shape During ImageNet Training?** Now we quantify the amount of shape encoded in the latent representations over the same ImageNet training snapshots as in Sec. 2.1. Fig. 5 shows the performance of the read-out module trained on the frozen SEN every five epochs, as well as the ResNet50's validation accuracy on ImageNet classification. Note that we train a separate read-out module for every snapshot. Similar to the findings in Fig. 3, we see that the majority of both binary and semantic shape information is learned within the first 10 epochs i.e., $\frac{77.1\%}{79.4\%} = 97.1\%$ of the final *Bin* mIoU and $\frac{52.9\%}{60.0\%} = 88.2\%$ of the final *Sem* mIoU is obtained by the read-out module after only 10 epochs. Contrasting purely shape related information (i.e., *Bin*), a small but significant portion of *per-pixel categorical* information is learned after the initial 10 epochs, when the learning rate decay is employed.

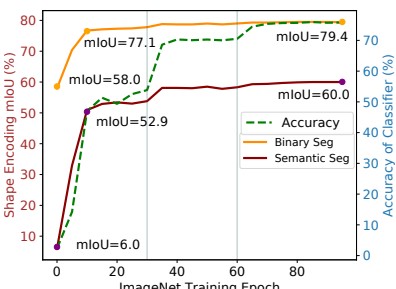

Figure 5: Quantifying the shape and semantic information encoded by a CNN over the course of ImageNet training. Vertical lines represent the learning rate decay.

### 3.2.1 DOES KNOWING AN OBJECT'S SHAPE IMPLY KNOWING ITS SEMANTIC CLASS?

We now explore whether a CNN encoding an object's shape necessarily implies that it also encodes the correct semantic category on a per-pixel level. In other words, for a frozen CNN, can a read-out module (trained for binary segmentation) successfully extract the binary mask while another read-out module (trained for semantic segmentation) cannot successfully extract the semantic segmentation mask? Previous results (e.g., Table 4, Table 5) show that, for certain layers and networks, the binary segmentation performance of a read-out module is much higher relative to the semantic segmentation performance. This suggests that shape information (i.e., the binary mask) and semantic information can be encoded in a *mutually exclusive* manner, i.e., a CNN can encode the silhouette of the object without encoding the semantic category of each pixel of the silhouette belongs to.

To this end, we validate various SENs and their read-out modules on stylized VOC12 *val* images as this ensures the networks must encode per-pixel semantic information based solely on the object's shape (note that stylization removes all texture information, see Sec. 2.2). The *difference* in performance between the *Bin* mIoU and *Sem* mIoU can therefore approximate the amount of shape information that is *not* correlated to its corresponding *semantic* class. As shown in Table 6, the large difference in performance between *Bin* and *Sem* suggests that these SENs capture the shape (i.e., *Bin* mask) of the object but lack the ability to correctly assign per-pixel semantic labels to these objects. Qualitative results are presented in Fig. 6; note how the binary mask (presented as likelihood heatmaps) for the SIN trained model reasonably segments the objects, while the semantic masks fail to resemble realistic predictions, i.e., *multiple* object categories are placed spuriously over the object of interest.

### 3.3 TARGETING SHAPE AND TEXTURE NEURONS

In Sec. 2, we used a dimensionality estimation technique to estimate the number of dimensions which encode shape and texture in a CNNs latent representations. Given these neurons, we now validate

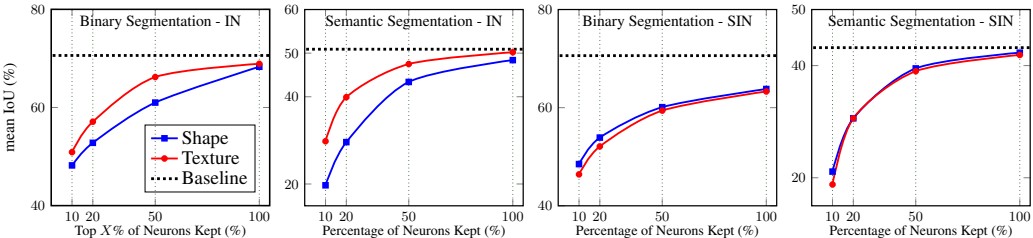

Figure 7: Shape encoding results by means of training a read-out module on the latent representations of an ImageNet (i.e., texture-biased model) (left two) and stylized ImageNet (shape-biased model)(right two) trained ResNet-50 for binary and semantic segmentation when removing all but top $X\%$ shape or texture-specific neurons.

if the most shape-specific, or texture-specific, neurons can influence the performance of a read-out module when keeping these specific neurons during training. We hypothesize that the network trained on Stylized ImageNet (i.e., a *shape biased model*) will be more reliant on the shape neurons than a network trained on ImageNet (i.e., a *texture biased model*) which are known to naturally exhibit a texture bias. See Appendix A.1 for additional experiments where we *remove* the targeted neurons instead of *keeping* them to further validate if the most shape or texture-specific neurons can influence the performance of a read-out module during inference. To asses this hypothesis, we conduct a series of read-out module experiment and the same settings as Sec. 3 are imposed. However, during this experiment we manipulate the latent representation as an image passes through the ResNet-50, before it is fed through the read-out module. More specifically, we rank the neurons by mutual information for both the *shape* and *texture* semantic factors, and then identify the top $X\%$ of neurons from either the shape or texture neurons. Then, we train read-out modules on the latent representations of two frozen ResNet-50s, one trained on ImageNet (IN) and another model trained on Stylized ImageNet (SIN). Before the latent representation is fed through the read-out module, we *remove all other neurons except for the top $X\%$ of shape or texture-specific neurons*. This forces the read-out modules to learn to perform binary segmentation and semantic segmentation solely from the top $X\%$ of neurons for either semantic factor, and we can identify which neurons are more heavily relied on for each network (i.e., the shape biased or texture biased model).

**Results.** Figure 7 illustrates the binary and semantic segmentation results in terms of mIoU obtained from training read-out modules on IN (left two) and SIN ((right two)) trained ResNet50s, respectively. It is clear that for the model biased towards texture (IN pretrained), keeping texture neurons while removing all other neurons results in a better performance than keeping only the shape neurons. In contrast, Fig. 7 (right two) shows that for shape-biased model (SIN pretrained), keeping shape-specific neurons achieves better performance than keeping only texture-specific neurons. These results support the hypothesis that the network trained on Stylized ImageNet (i.e., a shape biased model) is not only biased towards making predictions based on object shape, but more reliant on shape-specific neurons than a network trained on IN.

## 4 CONCLUSION

In this paper, we presented a systematic study of the capacity and quality of shape encoded in a CNNs latent representations. Approximating the mutual information between stylized PASCAL VOC images allowed us to estimate the dimensionality of the semantic concepts shape and texture (Sec. 2.1). We also designed a simple strategy for determining *how much* shape information is encoded in these latent representations, by training a read-out module on per-pixel binary segmentation ground truth labels. Additionally, we perform semantic segmentation to quantify how much of this shape encoding can be correctly attributed to per-pixel categorical information. We showed that a model pre-trained on ImageNet has weights that contain *almost* all the shape and categorical information needed to perform binary or semantic segmentation from the late stage features. We showed that CNNs encode a surprising amount of shape information at all stages of the network, but correctly assigning categorical labels to the corresponding shape only occurs at the last layers of the network, and that removing the image's texture information severely hurts this correspondence. Finally, we showed how removing all but a certain number of targeted shape or texture-specific neurons affects performance differently depending on the reliance on these neurons. These findings reveal important mechanisms which characterize a network's ability to encode shape information. We anticipate these findings will be valuable for designing more robust and trustworthy computer vision algorithms.

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

## A APPENDIX

### A.1 REMOVING SHAPE AND TEXTURE NEURONS

In Sec. 3.3, we performed an experiment where we kept the top $X\%$ shape and texture-specific neurons to compare how much different CNNs relied on these neurons to encode shape. We now perform a similar experiment, but instead of keeping the top $X\%$ neurons, we first identify the top $N$ of neurons from either the shape or texture-specific neurons. We then *remove these neurons* before passing the latent representation to the read-out module by simply setting the features at other dimensions to zero. This allows us to identify which neurons in each network (i.e., the shape biased or texture biased model) are relied on more heavily to encode shape and semantic information. Note that for this experiment, no training occurs. The goal is to simply measure the difference in inference performance when removing $N$ shape-specific neurons, or $N$ texture-specific neurons. Therefore we simply take the trained models and read-out modules from Sec. 3 to perform inference while masking out the targeted neurons. Note that validation is done on the *val* split from (non-stylized) VOC 2012.

**Experimental Details.** For this experiment, the dimensions sharing the most mutual information with respect to *shape* and *texture* are obtained from the same experiments from Sec. 2. We then rank the dimensions for each semantic factor by mutual information. Training and inference are done with the *trainug* and *val* split, respectively, from the (non-stylized) PASCAL VOC 2012 (Everingham et al., 2010) dataset.

Table 7: Shape encoding results for ResNet50's trained on ImageNet (IN) and stylized ImageNet (SIN) based read-out modules when the top $N$ *shape* or *texture*-specific neurons are removed from the latent representation during inference. Removing the top $N$ shape specific neurons from the SIN-read-out hurts the network's shape-recognition abilities more compared to the IN-read-out model.

| $N$ | Shape | | | | Texture | | | | Residual | | | |
|---|---|---|---|---|---|---|---|---|---|---|---|---|
| | **IN** | | **SIN** | | **IN** | | **SIN** | | **IN** | | **SIN** | |
| | *Bin* | *Sem* | *Bin* | *Sem* | *Bin* | *Sem* | *Bin* | *Sem* | *Bin* | *Sem* | *Bin* | *Sem* |
| **0** | **70.6** | **50.9** | **76.4** | **53.7** | **70.6** | **50.9** | **76.4** | **53.7** | **70.6** | **50.9** | **76.4** | **53.7** |
| **100** | 68.8 | 46.4 | 64.6 | 37.1 | 69.3 | 44.7 | 65.7 | 39.2 | 68.3 | 46.7 | 64.7 | 37.3 |
| **200** | 67.9 | 40.0 | 57.3 | 31.4 | 67.9 | 39.7 | 62.5 | 32.7 | 66.1 | 44.2 | 64.9 | 34.8 |
| **300** | 61.6 | 38.0 | 58.6 | 25.8 | 66.3 | 37.6 | 55.0 | 27.9 | 62.9 | 40.0 | 62.3 | 30.1 |

### A.1.1 RESULTS

Table 7 presents the binary and semantic segmentation results in terms of mIoU. We report the results under three different settings; (i) top $N$ *shape-specific* neurons removed, (ii) top $N$ *texture-specific* neurons removed, and (iii) top $N$ *residual* neurons removed. Note that for this experiment, we do not train the read-out module. Instead, we first remove the specified neurons, and then run inference using the pretrained IN and SIN models as well as the already trained read-out modules. Interestingly, we find that gradually removing the *shape-specific* neurons from SIN pretrained model more significantly hurts performance than the IN pretrained model. For instance, removing 100 shape-specific neurons from SIN achieves 37.1% *sem* mIoU, while the performance dropped to 25.8% *sem* IoU when the top 300 shape-specific neurons are removed (i.e., an 11.3% drop). When comparing this to the performance drop of the IN trained model, we see that the difference is lower, from 46.4% to 38.0% (i.e., an 8.4% drop). This further supports the hypothesis that SIN trained models are more reliant on the individual shape encoding neurons than the texture encoding neurons. In addition, removing shape-specific neurons from SIN pretrained model hurts performance more than removing the texture neurons. For example, when removing 300 shape neurons for the SIN trained model, the performance drops to 25.8%, while removing 300 texture-specific neurons decreases the performance to only 27.9%. Finally, we observe that removing *shape or texture* specific neurons hurts performance more than removing the residual neurons. This suggests that shape and texture are the two most important semantic factors for a network to encode and that other semantic factors contained in the residual (e.g., color, lighting) are not as discriminative for the task of image classification.

## A.2 Consistency of Dimensionality Estimation

Sec. 2.1 analyzed the consistency between the dimensionality estimate of (Esser et al., 2020) and that of (Bau et al., 2017). In order to quantify interpretability, the latter evaluates the alignment between neurons and semantic concepts using the Broden dataset, which consists of images with pixel-wise labelings of different semantic concepts. For each neuron $z_i$, it determines the top quantile level $T_i$ such that the activation value of $z_i$ exceeds $T_i$ only in $0.5\%$ of all observed cases over the dataset, i.e. $p(z_i > T_i) = 0.005$. Feature maps are then upsampled to the original image resolution and each neuron is thresholded according to its quantile $T_i$ to obtain a binary segmentation mask. A neuron is then termed a detector for a concept, if its segmentation mask has the highest intersection over union score (IoU) for this concept, and the IoU exceeds a threshold of $0.04$.

Because the Broden dataset contains no shape concepts, the comparison in Table 2 is limited to estimates on the number of neurons encoding texture, which is determined by the number of detectors for concepts from the categories material, texture and color of the Broden validation dataset.

We observe that both methods predict an inverse relationship between the estimated texture dimensionalities and the receptive field, except for the case of BagNet9, where a sudden drop in the number of texture detectors is observed for network dissection. Besides this qualitative agreement, the predicted absolute numbers differ. There are two main sources for the incompatibility in the absolute number of neurons. First, both approaches rely on hyperparameters,u i.e. the baseline score and the choice of the normalization function in the case of dimensionality estimation, and the chosen quantile and IoU threshold in the case of network dissection. Second, the semantic meaning of texture depends on the data, i.e. dimensionality estimation relies on the image pairs of SVOC, whereas network dissection relies on texture images of Broden. This might also explain the drop in detectors for BagNet9 if its receptive field is too small for some of the Broden textures. While absolute numbers depend on hyperparameters, results obtained with both methods are comparable across networks.

The dimensionality estimate relies on an estimate of mutual information from samples. This remains a challenging problem, and even powerful variational bounds exhibit either high-bias or high-variance and suffer from sensitivity to batch sizes (Poole et al., 2019). Besides statistical limitations on the ability to accurately estimate mutual information (McAllester & Stratos, 2020), even estimates which give neither upper nor lower bounds or those which give loose bounds are still useful in practice. For example, (Tschannen et al., 2020) demonstrate that loose bounds can lead to better representations when they are learned by mutual information maximization. For dimensionality estimation, potential biases of estimates will cancel out when comparing them between shape and texture neurons, hence an estimate based on the correlation is a suitable and efficient choice for our purposes.

## A.3 Estimating Shape and Texture Dimensionality of Different Networks Trained on Stylized ImageNet

We further estimate the dimensionality of shape and texture semantic concepts of different networks in Table 8 to test the consistency of the results reported in Table 1 on different architectures. We run the dimensionality estimation experiment (see Sec. 2.1) on AlexNet (Krizhevsky et al., 2012) and VGG-16 (Simonyan & Zisserman, 2015), trained on IN and SIN. Consistent with the findings for ResNet50, Table 8 shows that training on SIN increases the number of dimensions encoding shape and concurrently decreases the number of dimensions encoding texture: **AlexNet-IN**: [*Shape*=729, *Texture*=1299], **AlexNet-SIN**: [1119, 870]. **VGG-16-IN:** [710, 1321], **VGG-16-SIN**: [1090, 879]. The dimensionality estimation is done on the final representation before the last linear layer for all networks.

Table 8: Comparison of shape bias and shape dimensionality for different networks.

| Network | IN | | | | SIN | | | |
|---|---|---|---|---|---|---|---|---|
| | Factor $|z_k|$ | | Bias | | Factor $|z_k|$ | | Bias | |
| | Shape | Texture | Shape | Texture | Shape | Texture | Shape | Texture |
| ResNet-50 | 14.1% | 40.2% | 22.1% | 77.9% | 26.2% | 23.3% | 81.0% | 19.0% |
| AlexNet | 18.0% | 30.6% | 42.9% | 57.1% | 26.0% | 21.5% | 75.5% | 24.5% |
| VGG-16 | 15.3% | 37.9% | 17.2% | 82.8% | 26.6% | 21.5% | 77.4% | 22.6% |

### A.4 ESTIMATING SHAPE AND TEXTURE DIMENSIONALITY OF SELF-ATTENTION NETWORKS

We also experiment with the recently proposed Self-Attention Networks (Zhao et al., 2020), which replace convolutional layers with self-attention layers. Three different depths of Self-Attention Networks (SANs) were proposed. For fair comparison, we experiment with SAN-19, which the authors claim is the most similar size to ResNet50, in terms of the number of parameters in the network. Additional, SANs come with two types of layer operations, patch-based and pair-based. The patch-based SAN compares patches of pixels within the attention operations, while the pair-based SAN compares individual pixels and achieves lower performance on ImageNet. Due to the lower effective receptive field of the pair-based SAN compared to the patch-based SAN, we expect to see a larger number of neurons encoding shape in the patch-based SAN. The results are shown in Table 9. The patch-based SAN19 has the largest number of shape encoding dimensions and lowest number of texture encoding dimensions, when compared to the pair-based SAN19 and ResNet50.

Table 9: Comparing the number of shape encoding neurons and texture encoding neurons for self-attention networks (Zhao et al., 2020).

| Model | Factor $|z_k|$ | | Factor $|z_k|/|z|$ | |
|---|---|---|---|---|
| | Shape | Texture | Shape | Texture |
| ResNet50 | 349 | 692 | 17.0% | 33.8% |
| SAN-19 (patch) | 384 | 610 | 18.8% | 29.8% |
| SAN-19 (pair) | 304 | 764 | 14.9% | 37.3% |

### A.5 LAYER-WISE DIMENSIONALITY ESTIMATION ON ALEXNET

We now explore *where* another CNN encodes shape and texture at each layer of the network. More specifically, we apply the dimensionality estimation technique from Sec. 2.1 on AlexNet (Krizhevsky et al., 2012) on a number of different layers. Due to the different dimensions at each of these stages, we present the results as the percentage of dimensions encoding the particular semantic factor, $|z_k|/|z|$, where $|z|$ refers to length of the latent representation. The results are presented in Table 10 and Fig. 8. Note that the output from the convolutional layers also include the ReLU activation function.

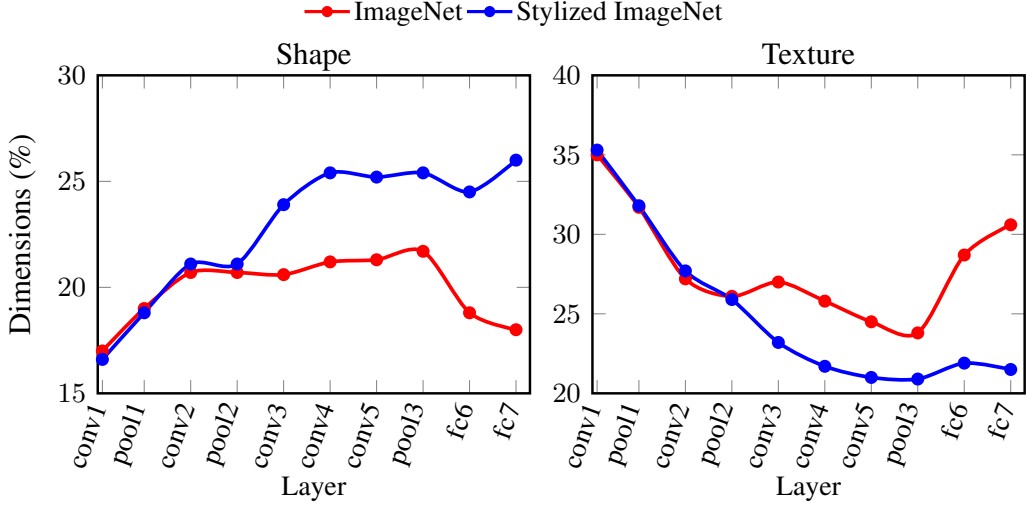

Figure 8: Shape (left) and texture (right) encoding dimensions estimated on each layer of AlexNet (Krizhevsky et al., 2012). Shape biased AlexNet trained on Stylized ImageNet (Geirhos et al., 2018) encode more shape at the later layers of the network which is consistent with the findings for ResNets (He et al., 2016).

Table 10: Percentage of neurons ($|z_k|/|z|$) encoding different semantic concepts, $k$, for different stages of **AlexNet** (Krizhevsky et al., 2012) trained for various levels of shape bias.

| Stage | IN | | SIN | |
|---|---|---|---|---|
| | Factor $|z_k|/|z|$ | | Factor $|z_k|/|z|$ | |
| | Shape | Texture | Shape | Texture |
| conv1 | 17.0% | 35.0% | 16.6% | 35.3% |
| pool1 | 19.0% | 31.7% | 18.8% | 31.8% |
| conv2 | 20.7% | 27.2% | 21.1% | 27.7% |
| pool2 | 20.7% | 26.1% | 21.1% | 25.9% |
| conv3 | 20.6% | 27.0% | 23.9% | 23.2% |
| conv4 | 21.2% | 25.8% | 25.4% | 21.7% |
| conv5 | 21.3% | 24.5% | 25.2% | 21.0% |
| pool3 | 21.7% | 23.8% | 25.4% | 20.9% |
| fc6 | 18.8% | 28.7% | 24.5% | 21.9% |
| fc7 | 18.0% | 30.6% | 26.0% | 21.5% |

