# OpenReview forum: "Shape or Texture: Understanding Discriminative Features in CNNs"
_ICLR.cc/2021/Conference — ICLR 2021 Poster_

### Official Review · AnonReviewer1 · 2020-10-25
**Contribution limited and results not interesting**

**Rating:** 4
**Confidence:** 4

**Review:**

This paper performs a detailed analysis of shape vs texture tradeoff in deep neural networks. The authors use two methods to analyze the learned shape features in CNNs: one based on estimation of concept dimensionality, one based on reconstructing shape from latent representations.

1. The contributions of this work is limited. First of all, the methods of this work is adopted or based on previous methods. The dimensionality estimation method is based on (Esser et al, 2020). The idea of using “read-out module” to segment images is similar to the idea in (Hermann et al, 2019) where a linear classifier is trained to predict the shape label.

    Moreover, the results are not interesting. The analysis conclusions made in this paper are all expected and not surprising, though i have to acknowledge that this is a more fine-grained study. I would suggest the authors to stress a few key messages that you are trying to make in this paper, rather than the lots of analysis results presented in the current one without a central conclusion.

2. Overall, I am not satisfied with the structure of this paper. It seems that the authors   struggle to put two not so relevant ideas together. I would like the authors to spend some time to discuss the motivation for using the two analysis methods and the dependencies between them. This will make this work coherent and consistent.

3. The design of readout module is not clear to me. Specifically, why did the authors choose the readout module to be “either one or three convolution layers with 3×3 kernels”? The current one seems to be too simple, can their be other designs for the read out module?

4. What hidden layer is analyzed for CNN in Table 1?

5. What exactly is shown in Figure 1, i understand the subfigure “stylized image” and “semantic”, but what are the two figures in the middle “GT” and “Shape”? What are these two images? What does GT means? I didn’t find the authors rigorously define these for figure 1 in the paper.

After rebuttal:
I appreciate the authors for the response. However, I feel that the conclusions drawn from this paper are not focused, there is no central message that i can get as a clear take-away for understanding CNNs after reading the paper. I do agree with each of the findings the authors make, it is just that the presentations and writings of the paper make me confused about what the authors are trying to convey through this paper. Therefore, i am keeping my score.

---

> ### Author Response · Authors · 2020-11-16
> **Author Response to Reviewer 1**
>
> Dear Reviewer 1,
>
> Thank you very much for reviewing our paper. We provide a point-by-point response to each concern below.
> 1. We believe AR1’s concerns regarding the degree of contribution may be due to a misunderstanding of our contributions. We do not claim that the techniques [2,3] used in our work are our contributions. Rather, we use these as tools in our empirical study to address important and remaining open questions (agreed on by AR2, AR3) on the ability of CNNs to encode shape information. We choose these tools to address issues that arise with previous studies [1,2]: [1] provides no solution to predict the number of ‘shape’ encoding neurons, and is limited to textures. [2] can correctly classify the a stylized image without encoding the full object’s shape (see Fig 6). Therefore, [2] cannot answer a question of focus in our paper: ‘What fraction of the object's shape is actually encoded in the latent representation?’. As agreed on by AR3, we extend beyond [2] by investigating the internal representations of shape vs texture on a per-pixel level rather than solely the external performance characteristics on image classification. Therefore, we believe that the methods used in our work have advantages when compared to [2] and we reveal several novel findings which are of interest to the CV/ML community. Note that the main contributions and key messages are laid out in Sec. 1.
>
>    We respectfully disagree that the results found in our paper are ‘not interesting’, ‘not surprising’, and are ‘all expected’. Other reviewers disagree with this assessment as well. E.g., AR3: “The sharp increase in shape representation dimension in the last stage of resnet is an interesting finding, as is the rapid growth of shape representations early in training.” AR2: “Fig. 3 shows an interesting plot”. Since the review does not explain why our results are not ‘interesting’ or ‘surprising’, we summarize what we believe to be important findings of our ‘fine-grained study’:
> - The majority of shape information is learned in the first 10-20% of epochs, suggesting that the number of shape neurons are quickly learned, and the following 80% of epochs involve optimizing the coding within these neurons.
> - The sharp increase in the amount of shape encoding in the last stage of a ResNet suggests that a fruitful avenue of research may be to guide CNNs to encode more shape at earlier stages.
> - Shape-biased models can correctly label a stylized image without encoding the full object's shape and the associated semantic categories. This reveals a weakness of [2] as their method attributes a perfect shape encoding for every correct prediction even if the shape encoded is a fraction of the total object's silhouette. We believe revealing a non-trivial weakness in a well cited method is intrinsically interesting to the research community who have used this approach in their work.
>
>    We welcome feedback on our assessment of these findings.
>
> 2. Thanks for the suggestion to discuss the motivation clearly. We believe that the two ideas are relevant and highly related: Alg. 1 allows us to identify which neurons fire specifically for the abstract semantic concepts of shape and texture. Alg. 2 then gives us a method of evaluating the amount, and effectiveness, of the shape encoding contained within the latent representations. With the intent of better demonstrating the relationship between the two sections, we have included two new experiments in Sec. 3.3 and A.1 of the revised manuscript based on suggestions from AR3. Please see the following thread for details: https://openreview.net/forum?id=NcFEZOi-rLa&noteId=kCz30gFYBSY.
>
> 3. We choose a low capacity read-out module as the objective is to evaluate the backbone’s ability to encode shape information. Using a read-out module that is too strong may be able to extract shape information from spurious correlations within the internal representations and will not fairly compare the shape encoding from different backbones.
> 4. We use the last layer before the fully connected layer (i.e., Stage-5), with a latent dimension size of 2048. We have updated the Tab. 1 caption in the revised paper.
> 5. A ResNet50 trained on Stylized ImageNet (i.e., shape biased) correctly predicts the image label (bird). However, extracting binary (3rd col.) and semantic (4th col.) segmentation maps with a one convolutional layer read-out module shows that it fails to encode the full object shape and fails to categorically assign every object pixel to the ‘bird’ class. ‘GT’ refers to the semantic segmentation ground-truth and ‘shape’ denotes the binary segmentation map. We have updated the caption of Fig. 1 in the revised paper.
>
> Again, thank you very much for your review. We would appreciate it if you could let us know whether this clarifies your concerns and changes your assessment of how interesting the findings of our work are as well as the contributions of our work.
>
> [1] (Bau, 2017)
> [2] (Hermann, 2020)
> [3] (Esser, 2020)

---

> ### Author Response · Authors · 2020-11-21
> **Request for Discussion**
>
> Dear Reviewer 1,
>
> We believe that we’ve addressed all of your concerns in the response below and revised paper draft. Please let us know if there are any remaining concerns or questions! We would especially appreciate a response before the discussion period ends on Tuesday so that we can clarify any further questions.

---

### Official Review · AnonReviewer4 · 2020-10-27
**Comprehensive analysis but limited novelty**

**Rating:** 4
**Confidence:** 4

**Review:**

The method provides two measures for assessing the degree at which shape is represented in CNNs. The first measure attempts to asses, within a given representation layer, the dimentionality required to encoder shape information. The second, evaluates the per-pixel shape representation, by attempting to generate the input's segmentation mask from the given representation (activation of the image at that layer).

*****

Pros:

The method provides a comprehensive review of the encoded shape information, both on the representation level and pixel level.
It answers questions in a variety of settings:
1. The shape encoded in different stages, or representation layers, for standard networks, and for networks trained to aleviate the shape bias (SIN, SIN + IN)
2. The degree of shape representation as a function of epochs or time trained.
3. The per pixel representation of shape, i.e to what degree can the semantic map of the input image be recovered from a given representation layer.

The paper is also very clear and results are presented well and well explained.

Cons:

Regarding the first of the two methods (A in Fig 2), I am not convinced of its correctness. Eq. 1 upper bounds the mutual information using variable p_i, which can be calculated. However, with a lack of a lower bound, this doesn't really mean much. The actual mutual information can be much higher. Why is this bound tight?

Regarding the second of the algorithm (B in Fig 2), existing algorithms in literature, presented for example in [1], learn a classifier, that given the representation of an image, attempts to classify it shape. As such novelty of Alg.2 is very limited: Instead of classifying the shape, the method is trained to produce a segmentation map.

In addition, the use of a classifier as in [1], could also be used to quantify the use of shape in different representation layers. Why is it better to use the existing formulation instead of a classifier? Can't the use of a classifier also replace the need of Alg.1 in the sense of capturing the amount of shape information used? Does the existing measure reveal something more interesting in Tables 1-4 and Figure 3?

********

Overall, while the paper provides a comprehensive analysis of the use of shape in CNNs, both on pixel level and the representation level, I am not sure of the correctness of Alg.1 and the originality and relevance of Alg.2 given existing work.

[1] The Origins and Prevalence of Texture Bias in Convolutional Neural Networks. Hermann et al.,

---

> ### Author Response · Authors · 2020-11-15
> **Author Response to Reviewer 4**
>
> Dear Reviewer 4,
>
> Thank you very much for reviewing our paper and the positive comments regarding our work; “a comprehensive review of the encoded shape information, both on the representation level and on the pixel level”, “the paper is also very clear and results are presented well and well explained”. We provide a point-by-point response to your concerns below.
>
> Correctness of Eq. 1
>
> AR4 raised a concern regarding an upper bound on the mutual information (MI) in Eq. 1 and lack of a lower bound; however, Eq. 1 is a lower bound on the MI, which has been shown in previous work (Eq. 11 in Foster & Grassberger, 2011). In the original paper (Sec. 2.1), we stated both the conditions under which this bound holds (both marginal distributions are Gaussian), and the conditions under which it becomes tight (jointly Gaussian). But in general, addressing the issue of the tightness of the bound, we do not need an estimate of the absolute MI for our purpose. Only the differences between the MI for shape and MI for texture pairs are used to quantify the amount of shape and texture-specific neurons. Even if we underestimate the true MI for shape and texture, the relative difference is meaningful and only the absolute numbers might not be as we have also discussed in A.1 in the original paper (now A.2). We thank AR4 for making us aware of the need for further clarification regarding MI estimates. Please see Sec. A.1 (now A.2) for further discussion in the revised paper.
>
> Novelty of Alg. 2
>
> We believe AR4’s concern regarding the novelty of Alg. 2 may be due to a misunderstanding of our contributions. We do not claim that Alg. 2 is a novel method, we use it to perform an empirical study on the ability of CNNs to encode shape information in a per-pixel manner, something that has not been sufficiently studied previously. Our use of Alg. 2 allows us to answer our unique questions: ‘What fraction of the object's shape is actually encoded in the latent representation?’ and ‘Does the network classify shape from only partial shape?’. [1] use a linear read-out classifier to classify the object category of the boundary contained in the image, which may not consider the entire object shape. This is a key finding from our work: just because a network can classify the object shape, does not imply that the entire object's shape is encoded in the network’s latent representations (see Fig. 1). Therefore, we strongly believe that our approach (Alg. 2) is a strictly more fine-grained method for robustly quantifying the amount of shape information contained in latent representations than previous work [1]. As an aside, linear read-out modules are used in many previous works as a tool to evaluate learned representations.
>
> Use of a classifier as in [1] compared with Alg. 1 in our work
>
> Alg. 1 has the following advantages over the strategy in [1]:
> - Alg. 1 is a more fine-grained quantification method than the linear classifier approach from [1]. This is because it estimates the dimensionality of a semantic factor (e.g., in the range [0,2048]) given a single image pair while the linear classifier approach will associate a 100% accuracy for an image as long as the shape-label is correctly predicted.
> - [1] does not control for the linear classifier predicting the correct class from only partial shape (e.g., see Fig. 1).
> - The metric used in [1] does not estimate the number of shape-specific dimensions. To show how estimating the shape-specific dimensions (Alg. 1) can be leveraged in a way that a linear classifier method cannot, we conduct experiments which are discussed in this thread: https://openreview.net/forum?id=NcFEZOi-rLa&noteId=kCz30gFYBSY
>
> Does the existing measure reveal something more interesting in Tables 1-4 and Fig. 3?
>
> We apologize but this question is slightly ambiguous to us and welcome clarification if we have misinterpreted it. Our interpretation is “Does Alg. 1 reveal something more interesting in Tables 1-4 and Figure 3 than previous work?”. We believe Alg. 1 produces the following interesting and novel findings:
> - Tab. 3 reveals a significant increase in shape encoding neurons at the last stage of the network. We also show that biasing a CNN towards shape mainly affects the last stage. We believe these are novel and interesting findings, as does AR3.
> - The majority of shape is learned during the first 10-20% of training epochs (Fig. 3). This suggests that the number of neurons which learn shape are quickly learned, and the following 80% of training epochs involve optimizing the coding within these neurons.
> - The proportion of neurons which encode shape compared to texture in a shape biased model is an interesting finding (Tab. 1(b)) and, to our knowledge, has not been previously analyzed.
>
> Thank you once again for reviewing our paper. We would appreciate it if you could let us know whether this clarifies your concerns regarding Alg. 1 and Alg. 2, and changes your assessment of the novelty of our work.

---

> > ### Comment · AnonReviewer4 · 2020-11-22
> > **Thanks for the clarification**
> >
> > Thank you for the detailed comments and clarifications made. The explanation regarding MI is now clear. I would suggest to clarify that novelty of the paper is to "perform an empirical study on the ability of CNNs to encode shape information in a per-pixel manner".
> >
> > I am still concerned about the need for Alg.1 and Alg.2 to draw the conclusions made. Specifically, you mention:
> > - "Tab. 3 reveals a significant increase in shape encoding neurons at the last stage of the network. We also show that biasing a CNN towards shape mainly affects the last stage.". If I were to use the classifier presented in [1], would I not arrive to the same conclusion? Specifically, If I were to use [1] for every stage of the network, I expect to see an increase in classification ability of shape, towards the last stage, from which I would also be able to draw that "biasing a CNN towards shape mainly affects the last stage".
> >
> > - "The majority of shape is learned during the first 10-20% of training epochs (Fig. 3)". Once again, I could use the technique of [1] for different epochs of training, and observe the change in classification ability of shape. I would expect, that a similar conclusion could be made here as well.
> >
> > - "The proportion of neurons which encode shape compared to texture in a shape biased model is an interesting finding (Tab. 1(b)) " - Why is this an interesting finding, beyond what [1] tells me about shape encoded in different layers? If I were to design an algorithm which tackles the imbalance between shape and texture, would't reducing the bias to shape, according to [1]'s classifier, be sufficient?

---

> > > ### Author Response · Authors · 2020-11-23
> > > **Clarification Concerning the Need for Algorithms 1 and 2**
> > >
> > > Dear Reviewer 4,
> > >
> > > Thank you for the response. We appreciate the suggestion to make the novelty of our paper more clear. We have updated the first sentence of the `Contributions’ accordingly in the revised manuscript.
> > >
> > > Necessity of Alg. 1 and Alg. 2:
> > > -
> > > Since our contributions are the findings and not the algorithms, the possibility there exists other algorithms which can be used to come to similar conclusions as in our paper does not reduce the novelty of our work. We do not claim that other methods (existing or non-existing) could not reveal similar findings as in our paper. Similar to how [1] used an existing algorithm (linear classifier) to analyze the shape vs texture bias, we pick two different algorithms to use in our study to reveal novel findings, which we have provided the following motivations for choosing:
> > >
> > > - Alg. 1:
> > >
> > > Alg. 1 has the following advantages over the strategy in [1]:
> > >
> > > 1. Alg. 1 is a more fine-grained quantification method than the linear classifier approach from [1]. This is because it estimates the dimensionality of a semantic factor (e.g., in the range [0,2048]) given a single image pair while the linear classifier approach will associate a 100% accuracy for an image as long as the shape-label is correctly predicted.
> > >
> > > 2. [1] does not control for the linear classifier predicting the correct class from only partial shape (e.g., see Fig. 1).
> > >
> > > 3. The metric used in [1] does not estimate the number of shape-specific dimensions. To show how estimating the shape-specific dimensions (Alg. 1) can be leveraged in a way that a linear classifier method cannot, please see Sec. 3.3 and Appendix A.1 in the revised paper.
> > >
> > > - Alg. 2:
> > >
> > > [1] cannot answer a question of focus in our paper: ‘What fraction of the object's shape is actually encoded in the latent representation?’. Thus, we choose Alg. 2 to quantify the percentage of shape encoded for a single image for both binary shape and semantics. We reveal that even shape biased networks do not encode the full object shape in the latent representations, and do not successfully associate the correct semantic category with every object pixel (see Fig. 1). This reveals a weakness of [1] as their method attributes a perfect shape encoding for every correct prediction, even if the shape encoded is a fraction of the total object's silhouette.
> > >
> > >
> > > Layer-wise comparison of shape encoding with [1]
> > > ----
> > >
> > > We agree with AR4 that we also expect an increase in classification ability of shape, towards the last stage of the network; however, Fig. C.4 in [1] shows contrasting results. Fig C.4 shows the linear classifier performance on shape and texture labels (i.e., from Stylized ImageNet) from AlexNet layers. They find that shape is the most decodable in the early layers and decreases subsequently after the `conv4’ layer. Further, it shows that the conv1 layer has more shape information than conv5. Conversely, we also show that shape is encoded and decodable in the previous layers, but not nearly as much as in the late layers. To further show the consistency of Alg. 1, we run it on each layer of AlexNets trained on IN and SIN to compare with [1] (Fig. C.4). We have revised the paper by including the results of this experiment in Appendix A.5 (Fig.8 and Table 10). From this experiment, we conclude that the early layers have less shape than later layers and that SIN trained AlexNet has more shape in the later layers of the network, even the fc layers. These results are consistent with results from Table 3 and your intuition.
> > >
> > > Why is the number of shape and texture neurons interesting or useful
> > > -----
> > >
> > > One motivation of using Alg. 1 over [1] is the unique ability of Alg. 1 to identify the shape and texture-specific neurons. Identifying these neurons enables the ability to target these neurons for a variety of downstream tasks]. The first and most salient example of this is the new experiments done which remove and keep targeted shape and texture neurons (please see Sec. 3.3 and Appendix A.1 for a detailed description). As future work, the most direct extension of our findings would be to maximize the number of shape encoding neurons as the mutual information approximation is fully differentiable with respect to the network's parameters. Maximizing this as an objective function could potentially increase the ability of the network to encode shape information and improve overall performance.
> > >
> > > Shape encoding over training epochs
> > > ----
> > >
> > > Again, the method of [1] can be used to analyze a network over training; however, we believe our motivations for choosing our algorithms still provide more meaningful results for this experiment.
> > >
> > > Thank you for the response and further discussion! Please let us know if we have addressed your concerns regarding the motivation of using Algs. 1 and 2 and if you have any more questions or concerns.

---

> ### Author Response · Authors · 2020-11-21
> **Request for Discussion**
>
> Dear Reviewer 4,
>
> We believe that we’ve addressed all of your concerns in the response below and revised paper draft. Please let us know if there are any remaining concerns or questions! We would especially appreciate a response before the discussion period ends on Tuesday so that we can clarify any further questions.

---

### Official Review · AnonReviewer2 · 2020-10-28
**New approach to understand the shape bias as a function of the visual hierarchy of CNNs**

**Rating:** 7
**Confidence:** 4

**Review:**

Pro’s:
* The question that the authors are trying to understand is interesting and relevant in the field of object recognition, texture/shape bias, and learned representations in deep neural networks.
* Authors provide a nice set of controlled experiments that suggest some of these effects, and authors present a scientific login in their approach (that although it is not perfect), it is quite relevant for the field. This is not “yet another paper on trying to overcome the texture bias without any intuition what-so-ever, to try to get better numbers (as has unfortunately been done in computer vision these days)”, on the contrary, this paper is about understanding the underlying mechanisms of the texture/shape bias that extend the final stages of computation in the visual hierarchy, and is why I am leaning towards accept.
Nearly all figures in the paper are clear and help convey what the authors are trying to express (although see somes clarification points)
* The paper is clear and easy to read/understand.

Con’s:
* Some experimental evaluations/methods are not clear (see below Observeration/Clarifications that if clarified may convince me to raise my score).

----

Observations/Clarifications:

Figure 3 shows an interesting plot. It would have been great to compare the result with that of Stylized ImageNet as well (that presumably will have a greater increase in shape-tuned neurons vs texture-tuned neurons).

What is a “stage” in the ResNet? Is this a residual block, or a specific layer? Stages are mentioned and used throughout the paper but are never formally defined. I have an intuition that they are related to the depth in the hierarchy of the network, but not on the layer type or the computation (example: is it a conv layer, is it rectified?) That being said, why is Stage 1 not included at all? (presumably it is the input/image so it is redundant?) Authors should make this more clear.

The Binary and Semantic Segmentation tasks are not clear. The binary segmentation task’s goal is to purely create a binary mask of any object? Or more than one? Does the mask have to be a closed contour? Further what is confusing about the semantic segmentation task is if the pixel-wise labeling is done GIVEN the ground-truth binary mask or the predicted binary mask. This should be made more clear in section 3.1

Section 3.2.1: Knowing an Object’s shape implies knowing its semantic class: I don’t think the results in Table 6 actually show what the authors claim. If by default the output of the binary masks are different, these effects will naturally carry over into the semantic segmentation procedure. The only way to truly account for the proposed question of “if shapes implies better semantic class”, is if all systems are equalized and given with the SAME binary mask (assume ground truth), and must use this oracle to then perform semantic segmentation. If the disparity is still great, then the before-mentioned claim has stronger computational support. Otherwise, I can see that there the confounding variable is that the nature of one system knowing how to binarize the image better a priori, will lead to differences in the final semantic segmentation score/maps.

----

Altogether the paper is well written and there is a good body of results/evaluations that support some of the claims the authors are doing. The problem they are working on is interesting and relevant to the ICLR and greater CV/ML/Vision Science community. I am leaning towards acceptance, and would appreciate it if authors clarify my doubts above to raise my score.

----
*Update as of December 5th. I would like to thank the authors for clearing up my concerns and would like to raise my score from 6 to 7. I think this paper would be a good poster that can provide complimentary insights through different experiments to the recent paper of Hermann et al. NeurIPS 2020.

---

> ### Author Response · Authors · 2020-11-15
> **Author Response to Reviewer 2**
>
> Dear Reviewer 2,
>
> Thank you very much for your valuable feedback. We appreciate your assessment of our work as a “not yet another paper on trying to overcome the texture bias without any intuition what-so-ever, to try to get better numbers” and that the problem we are working on “is interesting and relevant to the ICLR and greater CV/ML/Vision Science community”. We address your suggestions and questions in a point-by-point response below.
>
> Figure 3 for Stylized ImageNet
>
> We agree with you that reproducing the experiment summarized in Fig. 3 with the Stylized ImageNet (SIN) method would be interesting to compare the result with that of ImageNet (IN). As suggested by AR2, we first generate the stylized ImageNet using the publicly available code. We then train ResNet50 on SIN using the same set of training hyper-parameters for a fair comparison with the IN model. Then, for each checkpoint, we measure the dimensionality of the shape and texture semantic factors and plot the results in Fig. 3 (right) of the revised manuscript. Interestingly, as AR2 expected, the percentage of shape-tuned neurons grows much larger and faster than the IN trained model. This suggests that SIN provides a much stronger signal for the network to learn shape, particularly during the first few epochs of training.
>
> What is a “stage” in the ResNet?
>
> Stage refers to the feature map of a residual block within the ResNet (e.g., stage-5 refers to the feature map from the last residual block in ResNet). We have updated Section 2.2 in the revised manuscript to make this more clear.
>
> Why is Stage 1 not included?
>
> Given that the Stage-1 (f1) features in a ResNet correspond to the output of a single convolutional layer with stride 2, followed by a max-pooling layer, we did not include the result. However, we agree with you that even stage-1 features could be informative. As suggested by AR2, we ran the dimensionality estimation and read-out experiments using stage-1 features and now include the results in Table 3 and Table 5 of revised manuscript. From the results of these experiments with Stage-1 latent representations, we draw the following conclusions:
> Table 3: The first stage has the lowest percentage of shape neurons and highest percentage of texture neurons. Note that all three training procedures (i.e., IN, SIN, and (IN+SIN)->IN) have the same percentage of shape and texture neurons. This suggests that regardless of the training method (i.e., IN or SIN), it is difficult to encode more shape with only a single convolutional layer.
> Table 5: A small amount of shape information can be extracted from stage-1; however, these features lack high-level semantics to correlate with this shape information. Training a readout module by combining the features from all the stages (f1+f2+f3+f4+f5) in hypercolumn style does not improve the Bin performance, and lowers the Sem performance when comparing the performance to the other four (f2+f3+f4+f5). This suggests a single conv layer does not have the capacity to extract any shape or semantic information not already contained in the other stages.
>
> Clarifying the Binary and Semantic Segmentation tasks
>
> The goal of the binary segmentation task is to produce a binary mask of all objects in the image (i.e., one or more). The mask does not have to be a closed contour as there can be holes within the object masks or occlusions in the image.
>
> Please note that the semantic segmentation task does not take any binary mask as input whatsoever. The binary and semantic segmentation experiments are done completely independently of one another. To further clarify the training and validation process for the semantic segmentation task: Training - images are passed through a frozen CNN (e.g., ResNet50) and the latent representations are obtained (e.g., Bx2048xHxW dimensional feature maps at the last stage of a ResNet-50). The read-out module takes these latent representations as input and is trained to predict the semantic segmentation map of the input image (i.e., we only use the ground truth semantic segmentation mask during training as a supervisory signal). We appreciate the comment and updated Section 3.1 and Fig.2 (B) to better distinguish the difference between the binary and semantic segmentation tasks.
>
> Knowing an object’s shape implies knowing its semantic class
>
> We believe that the above explanation will help to alleviate any misunderstandings of the findings in Section 3.2.1. To reiterate, neither the groundtruth binary mask, nor the predicted binary mask, are used at any point to obtain the semantic segmentation prediction. The binary segmentation and semantic segmentation predictions are acquired completely independently of one another.
>
> Again, thank you very much for reviewing our paper and for your valuable suggestions! We would appreciate it if you could let us know whether our response clarified your concerns and if this changes your assessment of our work, or if there are any other questions you have.

---

> > ### Comment · AnonReviewer2 · 2020-11-24
> > **Resolved Comments + Quick follow-up**
> >
> > Dear Authors,
> >
> > Thank you for your detailed rebuttal and updated manuscript. All my doubts have been clarified, but I still have this looming doubt that even after I read Section 3.2.1 multiple times, I can't figure out if the interpretation of knowing an objects' shape implies knowing its semantic class (this is ideally the case), but I can't make sense of it in the updated version; There is a question in 3.2.1 in the subtitle that I am fond of, as I've suggested this!, but I can't make sense of the clear answer in the following paragraph. Having a shorter more digestible answer followed by a larger one that is less cluttered by hyperlinks to tables and graphs (potentially in the following paragraph) would be nice. I understand this is a complicated question with a non-trivial answer that if necessary may be further elaborated in the Supplementary Material.
> >
> > Thanks once again for the great effort in this updated version!

---

> > > ### Author Response · Authors · 2020-11-24
> > > **Revised Manuscript as Suggested by AR2**
> > >
> > > Dear Reviewer 2,
> > >
> > > Thank you very much for your response. We are happy that all of your doubts have been clarified. To address the readability of Sec. 3.2.1, we have updated the manuscript with a more concise answer which refers less to previous figures and tables.
> > >
> > >
> > > Our goal of Sec. 3.2.1 is to explore whether a CNN encoding an object's shape (i.e., binary segmentation mask) necessarily implies that it also encodes the correct semantic category on a per-pixel level (i.e., semantic segmentation mask). In other words, for a frozen CNN, can a read-out module (trained for binary segmentation) successfully extract the binary mask while another read-out module (trained for semantic segmentation) is unable to successfully extract the semantic segmentation mask? Prior to this experiment, we were expecting the shape-biased model (trained on Stylized ImageNet), which can correctly classify the image with the correct class label, to also encode the correct semantic category at each pixel for which it predicts an object resides. For example, the bottom row of Fig. 6 shows the read-out modules prediction for both binary (shape) and semantic (semantic) segmentation. If knowing an object’s shape implies knowing the semantic class, then the read-out module should be able to correctly extract the semantic segmentation map if the binary segmentation mask is reasonable. Surprisingly, this is not what we found. Even though the CNN and read-out module can easily localize the object of interest (i.e., the binary segmentation output of the cow), the network appears to think that the cow is made up of five different semantic categories (if you zoom in on the bottom right image of Fig. 6: person, dog, sheep, horse, cow). Note that the two read-out modules (i.e., for binary and semantic segmentation) are trained separately. Thus, our answer to the question in the Subsection header is `No’. Knowing an object’s shape does not imply knowing the semantic class.
> > >
> > > Again, thank you for the motivating comments and suggestions for further clarification. We would really appreciate it if you could let us know whether our clarifications and updates to the manuscript are enough to consider changing your rating.

---

> ### Author Response · Authors · 2020-11-21
> **Request for Discussion**
>
> Dear Reviewer 2,
>
> We believe that we’ve addressed all of your concerns in the response below and revised paper draft. Please let us know if there are any remaining concerns or questions! We would especially appreciate a response before the discussion period ends on Tuesday so that we can clarify any further questions.

---

### Official Review · AnonReviewer3 · 2020-10-29
**Interesting examination of shape vs texture in internal representations**

**Rating:** 8
**Confidence:** 3

**Review:**

Texture vs shape sensitivity is a basic and important question in understanding how deep convolutional nets work. This paper investigates this question by asking how CNNs represent shape versus texture information internally.  Using a (texture-vs-shape) stylized imagenet pioneered by Geirhos (2018), the paper applies a dimensionality estimation method of Esser (2020) and a segmentation-readout method for quantifying and visualizing the encoding of shape information in networks.  Several different network architectures and layers are compared; and results are compared to baselines from previous papers as well as natural lower- and upper-bounds.  The presentation is clear.

The paper makes several contributions, extending beyond Geirhos 2018 by investigating the internal representations of shape- vs texture rather than just the external performance characteristics.  It extends beyond Esser 2020 by more fully examining shape versus texture representations, and comparing a variety of networks and layers.  It extends beyond Bau 2017 by counting partially correlated neurons and explicitly training segmenters to explicitly read out local shape or class information.  The sharp increase in shape representation dimension in the last stage of resnet is an interesting finding, as is the rapid growth of shape representations early in training. The measurement of texture over shape representations in BagNet is a nice confirmation that the proposed measurements match observed behavior. The use of segmentation readout, tested several ways, provides helpful and new qualitative visualizations.

The paper could be strengthened if it proposed further testable hypotheses.  The paper does confirm that the measured shape dimensions is lower for BagNet architecture with reduced receptive field, but beyond observing correlations, the paper does not try to directly verify that the identified shape dimensions are responsible for the network’s shape-recognition abilities.  For example, would the shape segmentation readouts be more damaged by removal of the most shape-specific dimensions (vs other neurons)?  Or would the removal of these dimensions affect external shape or texture performance?  If an effect like this were verified, it would strengthen confidence in the proposed correlation-as-dimensionality measure of shape and texture content.

**edit, after revisions**  As mentioned in discussion below, I think the addition of intervention experiments makes the effects very clear and strengthen the paper.  I revised my score from 7 to 8.

---

> ### Author Response · Authors · 2020-11-15
> **Author Response to Reviewer 3**
>
> Dear Reviewer 3,
>
> Thank you for your review and feedback. We appreciate your assessment of our work as  “The presentation is clear”, “makes several contributions”,  as well as confirming that multiple findings from our work were “interesting” and a “nice confirmation”. We agree that this is an important area of research and thank the reviewer for succinctly summarizing the extensions beyond previous works (Bau 2017, Geirhos 2018, Esser 2020).
>
> "Would the shape segmentation read-outs be more damaged by removal of the most shape-specific dimensions (vs other neurons)?"
>
> We agree wholeheartedly with you that analyzing the performance of networks while targeting shape-specific dimensions would be a strong experiment to add to our paper. This experiment will also add an additional link between the efficacy of the two algorithms used in our paper (i.e., dimensionality estimation and read-out module). As suggested by AR3, we conduct two experiments while targeting either shape or texture specific neurons. One of the experiments evaluates the performance of the read-out module when removing the top N shape or texture specific neurons, and then second evaluates the performance of the read-out module when removing all but the X% of the top shape or texture specific neurons. We have revised the paper by including the results of these experiments in Sec. 3.3 and Appendix A.1, respectively.
>
> Please see the following thread we made for a more detailed description and findings from this experiment https://openreview.net/forum?id=NcFEZOi-rLa&noteId=kCz30gFYBSY.
>
> Thank you for the great experiment suggestion as we think these are important findings that can be leveraged in future work and, as suggested, make the current work stronger.

---

> ### Author Response · Authors · 2020-11-21
> **Request for Discussion**
>
> Dear Reviewer 3,
>
> We believe that we’ve addressed all of your concerns in the response below and revised paper draft. Please let us know if there are any remaining concerns or questions! We would especially appreciate a response before the discussion period ends on Tuesday so that we can clarify any further questions.

---

> > ### Comment · AnonReviewer3 · 2020-11-22
> > **New experiments strengthen the results**
> >
> > Thanks for the added experiments - especially A.1.  The appendix Figure 7 and 8 explicitly clarify how texture units are more essential to an ordinary network than shape units, and the how the situation is different (if not actually reversed) in a the SIN network.  Because the experiment in the appendix avoids double-negative reasoning, this reviewer found the results in the appendix easier to understand than the removal experiments in Table 7 in Sec 3.3; the authors may want to consider reversing which variant of the experiment is presented in the main paper.
> >
> > The revisions have strengthened the paper.

---

> > > ### Author Response · Authors · 2020-11-23
> > > **Revised Manuscript as Suggested by AR3**
> > >
> > > Dear Reviewer 3,
> > >
> > > Thank you very much for your response and again for the experimental suggestions which we agree has ‘strengthened the paper’. We also appreciate your additional suggestion of revising the draft and consider including the results from the new experiments in the main paper. As suggested by AR3, we have now reversed the results and have replaced the experiments from Sec. 3.3 with Appendix A.1.
> > >
> > > If you could let us know if these changes have improved your assessment of our work enough to consider changing your rating, we would greatly appreciate it.

---

### Author Response · Authors · 2020-11-15
**New Experiment: Targeting and Removing Shape and Texture Neurons**

Dear Reviewers,

Thank you very much for reviewing our paper. We have conducted additional experiments based off of the suggestions from AR3. We believe this will clearly demonstrate the meaningful connection that exists between the two algorithms used in our paper (AR1) and how they can be leveraged for future work: (i) dimensionality estimation of semantic factors via mutual information, and (ii) shape encoding quantification with convolutional read-out modules.

The main goal of these experiments is to evaluate the ability of a read-out module to extract shape and semantic information (i.e., binary and semantic segmentation) from a targeted set of either shape or texture-specific neurons. More specifically, we conduct two experiments after ranking each dimension by mutual information: (i) we perform inference (on PASCAL VOC 2012) with the already trained IN and SIN readout modules from Sec. 3, and simply remove the top N shape or texture neurons from the frozen encoder’s latent representation before feeding it to the read-out module. (ii) We train a read-out module for binary and semantic segmentation after removing all neurons but the top X% of the shape or texture-specific neurons from the frozen encoder’s latent representations. We have revised the paper by including the results of experiments (i) and (ii) in Appendix A.1 and Sec. 3.3 , respectively.

From this experiment we make the following four interesting findings:

Appendix A.1: Removing N shape-specific neurons from the SIN pretrained (shape-biased) model more significantly hurts performance than removing N shape-specific neurons from the IN pretrained (texture-biased) model. This supports the hypothesis that the network trained on Stylized ImageNet (i.e., a shape biased model) is more reliant on the shape neurons than a network trained on ImageNet (i.e., a texture biased model).

Appendix A.1: Removing N shape-specific neurons from the SIN pretrained model hurts performance more than removing N texture-specific neurons from the SIN pretrained model. This supports the hypothesis that a shape-biased network relies more on shape-specific neurons than texture-specific neurons.

Appendix A.1: Shape and texture are more discriminative semantic factors for image classification than residual factors (e.g., color) as the removal of N shape or texture-specific neurons hurts the read-out modules performance more than the removal of the top N residual neurons.

Sec. 3.3: Texture biased models (IN pretrained) are more reliant on texture-specific neurons, as keeping the top X% texture-specific neurons achieves better shape encoding performance (in terms of read-out mIoU on binary and semantic segmentation) than keeping the top X% shape-specific neurons.

For further results, details, and discussion, please see Sec. 3.3 and Appendix A.1 in the revised manuscript.

---

### Author Response · Authors · 2020-11-25
**Author's summary of rebuttal discussion and revision with suggested changes**

We thank all reviewers for their valuable feedback and we very much appreciate their assessment of our work as "interesting", "makes several contributions" (AR3), "not yet another paper on trying to overcome the texture bias without any intuition" (AR2), and "provides a comprehensive review of the encoded shape information" (AR4). We've updated our manuscript several times following great feedback and comments from the reviewers. This is a summary of changes, main concerns and how we addressed them.

- A new set of experiments (Targeting Shape and Texture Neurons) based off of the suggestions from AR3 to demonstrate the meaningful connection that exists between the two algorithms used in our paper. AR3 acknowledged ‘The revisions have strengthened the paper’ and indicated being happy with the updated version.

- We added a new plot in Fig. 3 as suggested by AR2 to analyze the number of dimensions in a ResNet50 that encode shape and texture over the course of Stylized ImageNet training. As expected, the percentage of shape-tuned neurons grows much larger and faster in the SIN trained model than the IN trained model. Additionally, we added stage-1 results in Table 3 and Table 5 as suggested by AR2. AR2 acknowledged ‘All my doubts have been clarified’ and indicated being happy with the updated version.

- Updated the writing of Sec. 3.2.1 to make it more clear as suggested by AR2.

- Added layer-wise shape encoding results for AlexNet to compare with (Herman, 2020) as suggested by AR4.

-  Added discussions to improve the clarity regarding the lower and upper bound in Eq. 1 as suggested by AR4.

- Novelty and Contributions (AR1, AR4): We addressed this point by clarifying that our contributions are the novel findings and not the algorithms. Note that we do not claim that the algorithms used are novel, but use these tools to perform an empirical study on the ability of CNNs to encode shape and semantic information in a per-pixel manner. We updated the `Contributions’ section in the introduction to make the novelty of our paper more clear as suggested by AR4.

- Further clarifications (AR1, AR2, AR4), the necessity of Algorithms 1 and 2 (AR4), Fig. 1 caption improvements (AR1), distinction between binary and semantic segmentation task (AR2) etc.: we have addressed all of them and updated the paper accordingly.

---

### Decision · Program_Chairs · 2021-01-07
**Final Decision**

**Decision:**

Accept (Poster)

**Comment:**

This paper proposes to do a fine-grained analysis of how shape and texture play a role in the decisions made by CNNs. Lots of recent evidence suggests that CNNs exhibit a texture bias, and there has been considerable effort in understanding where this comes from and how to overcome it. The paper focuses in particular on understanding: (a) what fraction of the neurons are devoted to shape-vs-texture (roughly speaking), and (b) per-pixel results using a convolutional readout function. The reviewers were divided at the time of submission and remained so at the end of discussion. At the end of discussion, the reviewers were split, with scores ranging from 4 (R1,R4), 7 (R2), and 8 (R3). The AC wants to thank and acknowledge the authors as well as all of the reviewers for their engagement in the discussion.

- R2 and R3 are largely positive, driven by the extent of the experiments and the number of interesting results (e.g., how the fraction of the dimensions used for shape changes as a function of depth in the network). Both had smaller non-critical concerns that were addressed (as far as the AC can tell) in the discussion.
- R4’s most important concern, in the AC's view, is the question: could these results / different conclusions have been obtained via linear probe methods like Hermann et al.? The authors argue that analyzing the fraction of neurons used and at a per-pixel is more fine-grained than linear probes. This boils down to an intangible question of contribution, on which the AC is inclined to agree with the authors and R2 and R3: analyzing the dimensions contribute provides, at least to the AC, a complementary view to the linear probe and that this will be of interest to the ICLR community (although see final comment). R4 also had a number of smaller concerns that seem to be largely addressed (e.g., about correctness).
- R1 argues primarily that the paper does not have a clear point or methodological contribution, for instance pointing out that readout modules were used in Hermann et al. or (as an example) arguing the readout function design is too simple.  The AC is inclined to agree with the authors’ response that the other reviewers seem to largely agree on the contribution (especially contributions via experiments rather than method) but disagree on how to weigh these contributions. The AC would also add that readout modules are a core idea for understanding neural representations that long predate Hermann and are by design (as the authors note) almost always as simple as possible.

At the end of the day, the AC is agrees with R2 and R3 for the contribution of the work and is inclined to accept. Given the other reviews, the AC does not agree with R1’s arguments, but would suggest that the authors think about how to sharpen their claims further. The AC is sympathetic to the concerns of R4, and urges the authors to think about a more concise and clean argument for R4’s concerns --- many other readers will have similar concerns and as clean of an illustration will be helpful. Overall, the AC believes that the paper’s methods, experiments and analysis are of interest and value and is thus in favor of acceptance.